# The Effects of Nordic Hamstring Exercise on Performance and Injury in the Lower Extremities: An Umbrella Review

**DOI:** 10.3390/healthcare12151462

**Published:** 2024-07-23

**Authors:** Hugo Nunes, Luís Gonçalves Fernandes, Pedro Nunes Martins, Ricardo Maia Ferreira

**Affiliations:** 1Polytechnic Institute of Maia, N2i, Social Sciences, Education and Sports School, Avenida Carlos de Oliveira Campos, Castêlo da Maia, 4475-690 Maia, Portugal; lfernandes@ipmaia.pt (L.G.F.); pmartins@ipmaia.pt (P.N.M.); 2Polytechnic Institute of Coimbra, Coimbra Health School, Scientific-Pedagogical Unit of Physiotherapy, Rua 5 de Outubro, São Martinho do Bispo, 3045-043 Coimbra, Portugal; 3Sport Physical Activity and Health Research & Innovation Center (SPRINT), 4960-320 Melgaço, Portugal

**Keywords:** nordic hamstrings, performance, injury

## Abstract

Due to their potential positive outcomes, hamstring eccentric exercises are becoming increasingly popular in training regimens. Among the various exercises, the Nordic Hamstring Exercise (NHE) is the most common. Despite its popularity, there are still some doubts about its benefits and risks. So, the aim of this umbrella review was to summarize the effects of NHE on performance and injury prevention. Following the PRISMA guidelines, a comprehensive literature search was conducted across multiple e-databases, according to the P (injured and non-injured athletes or recreationally active or healthy individuals) I (NHE) C (no intervention, placebo, or other interventions) O (performance or injury) S (systematic reviews) model. The quality of the studies was accessed with the AMSTAR-2. From the 916 systematic reviews found, only 10 could be included. They encompassed 125 studies, enrolling 17,260 subjects. The results from the studies indicate that NHE interventions demonstrated positive effects on sprint performance, muscle activation, eccentric strength, and muscle architecture (fascicle length, muscle thickness, and pennation angle). Furthermore, NHE is effective in preventing hamstring injuries (up to 51%). In conclusion, NHE should be integrated in training (especially, in the warm-up phase) for both enhancing athletic performance and preventing hamstring injuries. For achieving more positive results, it is recommended that high-volume is followed by low-volume maintenance, targeting 48 reps/week.

## 1. Introduction

The hamstring is a muscular complex made up of the biceps femoris muscle (long and short head), semi-tendinous and semi-membranous. Its function is the hip extension and the knee flexion [1]. Studies show that there is a variation in muscle activation depending on the type of exercise to be performed. The bicep femoris and semi-membranous are more activated in dominant hip exercises; in dominant knee exercises, the semi-tendinous muscle was the most activated [2]. Considering that each part of the hamstring muscle is anatomically and biomechanically distinct, different activation profiles are expected during exercises.

In the literature, there are several interventions that are intended to strengthen the hamstring and reduce its risk of injury (e.g., strain), such as Nordic hamstring exercise (NHE) [3,4,5,6]. The NHE is a focused eccentric-type exercise in which the individual, positioned on their knees, uses their hamstring muscles to resist a forward-falling motion [7]. It became popular in training regimens after the Mjølsnes et al., 2004 study [8]. It seems that NHE programs are effective in improving eccentric strength, muscle activation, and jump height performance [8,9,10,11], as well as reducing hamstring injury rates [11,12,13,14].

Eccentric training shows superiority over other types. Eccentric actions are more cost-effective (greater use of muscle strength with low energy cost) and cause a greater cross-training effect compared with, for example, concentric exercise [15]. In addition, eccentric training increases muscle size and strength [16,17], and when performing eccentric loading, it is expected to increase the physiological working length of the muscle, thus preventing sarcomeres from reaching a critical length [10].

Although several studies show us that NHE is beneficial for reducing the risk of injury, and is an excellent exercise to improve muscle performance, as well as for strengthening hamstring muscles [1,4,18,19,20,21], NHE continues to cause some doubts as to its applicability and reliability [7]. Furthermore, to our knowledge, no umbrella review has been conducted on the effects of the NHE on performance and injury prevention.

Therefore, the aim of this overview is to summarize, from systematic reviews, the effects of NHE on performance and injury.

## 2. Materials and Methods

In an attempt to ensure a high-quality study, this overview was conducted following the PRISMA (Preferred Reporting Items for Systematic reviews and Meta-Analyses principles) [22].

### 2.1. Search Strategy

In January 2024, systematic and comprehensive searches were conducted, aiming to identify systematic reviews that evaluated the effect off the Nordic exercises on performance and injury. The searches were conducted in the following electronic databases: PubMed, Research Gate, EBSCO, and B-ON. The study selection respected the PICOS (Patients, Intervention, Comparison, Outcomes, Studies) model to guide the search strategy (Figure 1):

For the search strategy, a conjunction of keywords, MeSH terms, and established search filters were used. The main umbrella terms used to search the databases were: “Nordic”; “Hamstring”; “Exercise”; “Eccentric”. The terms (and their associates/derivatives) were then combined with the appropriate truncation and Boolean connectors. 

An example of an online search strategy draft used in the Pubmed database is presented in Figure 2:

### 2.2. Study Selection Process

The reviewers independently screened the titles and abstracts yielded by the search against the inclusion and exclusion criteria, and performed the selection of the potential studies. 

Potential studies were compiled in EndNote (version 21), and the duplicates were removed using the automated software command “find duplicates”. Beyond this process, all the studies were manually reviewed to ensure that no duplicates remained. The authors then assessed the full-text versions and decided whether they actually met the eligible criteria. In cases where full versions were inaccessible or data were missing, the study’s authors were contacted. 

The study selection process was supervised, and the disagreements were solved through verbal discussion or arbitration by a third reviewer. The inclusion and exclusion criteria applied to this review are described in Table 1.

### 2.3. Data Extraction and Syntheses

Data collection and extraction were performed by one author, with another author verifying the process to enhance consistency. The selected study-associated documents (i.e., full document, supplementary material, appendices, and journal publications) were collected for analysis. The data that were extracted from the selected publications to assess the effects of Nordic exercise included the title, authors’ names, year of publication, participants’ sample size and their characteristics, objectives, description of the interventions, description of the control groups, studies’ outcomes, assessment times, studies’ results, and studies’ conclusions.

### 2.4. Outcomes 

Considering the broad scope of performance- and injury-related outcomes, it was decided to restrict the work to specific umbrella terms, such as strength, muscular activation, muscular architecture, speed, agility, range of motion, flexibility, balance, injury frequency and types, pain, and fatigue. Additionally, as secondary outcomes, this study also retrieved from studies any adverse effect found related to the NHE.

### 2.5. Quality Assessment 

The authors independently scored the bias of the studies by using the AMSTAR-2 [23]. The instrument is a critical appraisal tool for systematic reviews that has 16 items in total. The overall rating is based on weaknesses in critical domains: High (Zero or one non-critical weakness)—The systematic review provides an accurate and comprehensive summary of the results of the available studies that address the question of interest; Moderate (More than one non-critical weakness)—The systematic review has more than one weakness, but no critical flaws. It may provide an accurate summary of the results of the available studies that were included in the review; Low (One critical flaw with or without non-critical weaknesses)—The review has a critical flaw and may not provide an accurate and comprehensive summary of the available studies that address the question of interest; Critically Low (More than one critical flaw with or without non-critical weaknesses)—The review has more than one critical flaw and should not be relied on to provide an accurate and comprehensive summary of the available studies. 

## 3. Results

### 3.1. Selection of Studies 

A set of 916 records were identified via searching databases. After the application of the inclusion and exclusion criteria, 10 studies emerged. The flow diagram in Figure 3 summarizes the selection process.

### 3.2. Methodological Quality

The methodological quality assessment, using the AMSTAR-2, of the 10 selected papers [24,25,26,27,28,29,30,31,32,33] showed that the majority of the systematic reviews had a Critically Low classification (60%) [24,26,27,28,30,33]. The AMSTAR-2 identified three problematic items within the studies: the explanation for the selection of included study designs; accounting for risk of bias in individual studies when interpreting the results; and providing a satisfactory explanation and discussion of the observed heterogeneity. The item study selection in duplicate, and assessment of the risk of bias, were the least concerning. The classifications obtained are described in Table 2.

### 3.3. Study Characteristics

Overall, the 10 included systematic reviews [24,25,26,27,28,29,30,31,32,33] were published from 2008 [28] to 2021 [25,29,30,31,33] and conducted in in several countries: Australia [24,31], Belgium [27], Brazil [30], Canada [28], Qatar [32], Spain [25,29], UK [26,33].

The systematic reviews encompassed a total of 125 studies, with an average of 12.5 studies per systematic review (maximum: 29 [29]; minimum: 5 [24]).The majority of the studies included in these systematic reviews were RCTs-type studies (73%), conducted between 1998 and 2021 (2018 being the most common year—17%).

From the studies included in the systematic reviews, a total of 17,260 subjects were enrolled (mean: 1726 per systematic review; maximum: 8459 [32]; minimum: 181 [33]), predominantly male (86%), with ages ranging from 10 to 40 years. The majority of the sample consisted of athletes and recreationally active or healthy individuals, with soccer being the most commonly represented sport (71%).

Regarding the Nordic hamstrings exercise, the most common protocol involved 2–3 sets of 6–12 repetitions, with 30 s to 3 min of rest between sets, performed 1–3 times per week for a duration of 4–10 weeks. This exercise was often conducted either in isolation or incorporated into other workouts, such as strength training, FIFA 11 program, or regular warm-up routines.

For more detailed information, see Table 3.

## 4. Discussion

The main objective of this study was to thoroughly analyze whether NHE influences the performance and injury. Overall, the results of the systematic reviews analyzed suggest that the inclusion of NHE can have a significant positive impact on both injury prevention and athletic performance. The subsequent discussion will be structured to address the main outcomes observed, with an emphasis on aspects related to sports performance and injury incidence. Furthermore, whenever possible, practical guidelines will be provided for the effective implementation of these findings in the health and sports context.

### 4.1. Performance

Performance-related outcomes were the most explored among the included systematic reviews. In fact, 6 out of 10 reviews focused on these topics [25,26,27,29,30,33]. One of the most consistent findings across the studies was that the NHE has the potential to increase the knee flexors’ eccentric strength. However, NHE training protocols remain a topic of current interest and a challenge for coaches and conditioning trainers, as no consensus has yet been reached. In our study, the training protocols varied within the studies regarding volume, period, and weekly frequency. Balancing the NHE training dose to generate muscle adaptations without promoting muscular overload or excessive fatigue seems difficult. Nevertheless, a potential resolution is starting to emerge. Comparing different training volumes, it appears that higher training volumes have more capacity to enhance muscular strength [26,30]. For example, Severo-Silveira et al. (2018) [34] found that a progressive workload group (236 repetitions over 8 weeks) had better hamstring strength results compared to a constant workload group (138 repetitions over 8 weeks). However, due to restricted training schedules and overall routines, progressive NHE training during the season might not be suitable. A possible solution is to perform high-volume training followed by low-volume maintenance. Presland et al. (2018) [35] reported that two weeks of high-volume NHE training followed by four weeks of low-volume training have similar adaptations compared to six weeks of high-volume training (128 vs. 440 repetitions, over six weeks, respectively). Therefore, implementing high-volume training during the preseason and maintaining gains with low volume during the season may be more suitable for teams’ training routines.

Changes in eccentric strength are achieved via increased muscle excitability, influenced by muscle fiber types and the number, synchrony, and discharge of recruited motor units [27]. Among 12 exercises, the NHE showed the highest biceps femoris activation (>60%), especially the long head [29]. Beyond muscle excitability, muscle architecture also influences strength development. The effects of NHE on muscle architecture were one of the most explored across the studies. Muscle architecture influences both injury- and performance-related outcomes. Adaptations in muscle architecture are related to fascicle lengthening, pennation angle, and muscle thickness [27]. Evidence suggests that previously injured biceps femoris long head muscles have significantly shorter fascicles than those without an injury history [36]. This may increase the chance of sustaining a new injury, as biceps femoris long head fascicles shorter than 10.56 cm at preseason quadruples the risk of hamstring strain injury [37]. Furthermore, an 11% biceps femoris long head fascicle length increase can reduce hamstring strain injury probability by 21%, while a 50% increase in eccentric strength is required for a similar reduction [37]. One possible reason raised is muscles with shorter fascicles may be more susceptible to eccentrically induced microscopic damage when stretched, facilitating macroscopic damage [38]. 

Fascicle length increase is crucial for performance-related outcomes, impacting force–velocity and force–length relationships, directly affecting muscle function (longer fascicles may contain higher amounts of in-series aligned sarcomeres, which would increase muscle contraction velocity and prevent muscle damage due to over-lengthening) [39]. The NHE effectively increases fascicle length, positively impacting sprint performance [25]. This positive impact in the fascicle length is independent of the training periodization or participants’ conditioning status [25,30]. Both low- and high-volume NHE training can increase fascicle length [34,35]. For example, although effect sizes generally favored the group performing a higher training volume, it was found that a NHE performed once a week was equally effective at increasing fascicle length as a program performed two times per week (i.e., average weekly volume of 31 vs. 61 repetitions, for the one and two days/week groups, respectively). Collectively, these findings indicate that the NHE is a time-efficient intervention, with only approximately 48 repetitions per week being sufficient to improve the outcomes. This information is particularly relevant for clubs that have difficulties implementing high volumes of NHE training due to their tight training schedules. While fascicle length enhancement occurs from the first month of NHE training, this primarily applies to nonathletic populations [4]. Well-trained athletes require a longer training period (6–8 weeks) to see similar benefits [26,30]. Moreover, body mass can also influence results [25]. As NHE is usually carried out based on one’s own body height, heavier individuals may benefit more from the exercise by the increasing moment arm of this mass in relation to the knee and hip, which progressively increases the torque and the force requirements from the hamstrings, and consequently the intensity [26]. In contrast, for well-trained individuals with a low body mass, the exercise may not be sufficient to maximize adaptations [25]. Therefore, adding external loads may be required to stimulate continuous adaptations, particularly to lighter and more experienced athletes.

Muscle thickness and pennation angle also play roles in NHE effects [26,27]. They determine muscle strength by influencing the addition of in-parallel sarcomeres, enhancing maximum strength capacity [40]. However, increases in muscle thickness and pennation angle may also counter the tendency for fascicle length increase, thereby decreasing the muscle’s shortening capacity. It was found that largest pennation angle is associated with the lowest force relative to the muscle cross-sectional area, suggesting that excessive muscle hypertrophy could affect the muscle’s pennation angle and potentially limit fascicle lengthening and force production [41]. The systematic reviews showed that NHE could effectively decrease the pennation angle, enhancing their performance-related outcomes [26,27].

Is worth noting that NHE training generated significant increases in eccentric strength in all studies, regardless the type of evaluation performed. However, when tests were performed on the isokinetic dynamometer, strength gains ranged from 10 to 15%, whereas studies using the NHE device reported increases of 16–26% [30]. The greater strength gains reported with the NHE device are likely due to training specificity, as the NHE device mimics the exact movement performed during the training program, unlike the isokinetic dynamometry. Caution should also be taken when performing ultrasound assessment and extrapolation methods. Yagiz et al., 2021 [33] found that eccentric training has a small effect based on manual linear extrapolation (g = 0.29; 95% CI: −0.26, 0.85), a medium effect based on panoramic ultrasound (g = 0.72; 95% CI: 0.17, 1.28) and a large effect based on the trigonometric equation (g = 2.20; 95% CI: 0.99, 3.41). This highlights the importance for the scientific community to consider reaching a consensus for hamstring measurements to assess the impacts of training more accurately, thereby providing more comparable results between interventions.

### 4.2. Injuries 

Overall, the studies consistently showed that NHE could be an effective intervention for preventing hamstring injuries [24,28,32]. Results from the systematic reviews indicate a statistically significant and clinically meaningful reduction of up to 51% in hamstring injuries for athletes competing in various sports at various levels [24,32]. 

One relevant factor observed across the studies was poor adherence and high dropout rates, especially due to delayed-onset muscle soreness [28]. This is important, as other studies have found that eccentric strengthening with good compliance is associated with better hamstring injury prevention [42]. For example, it was found that when compliance rates were high (91%), NHE decreased the hamstring injuries [13]. Conversely, little benefit was achieved when compliance rates were low (21%) [43]. Therefore, to increase compliance and subsequently reduce injuries, NHE should be included in injury prevention programs, such as FIFA 11+ [24]. 

Understandably, most studies used the NHE as a warm-up exercise [24,31,32]. However, it seems that NHE could also be beneficial during the cool-down phase. It was found that performing eccentric hamstring strengthening exercises during the cool-down, rather than the warm-up, could maintain eccentric hamstring strength and preserve the functional strength ratio, even in a fatigued state [44]. This evidence underlines that a cool-down may increase muscle flexibility, and thereby assist in reducing muscle injuries and improving performance.

Another interesting finding was that pre-season knee flexor strength was not associated with hamstring strain injuries [31]. This finding was consistent regardless of whether strength was expressed as absolute, body mass-normalized, or as a limb-asymmetry percentage. Therefore, despite the widely acknowledged advantages of incorporating eccentric knee flexor exercises to mitigate hamstring injuries, eccentric knee flexor strength alone cannot predict their occurrence. This highlights the multifactorial nature of these injuries, suggesting that practitioners should consider other factors, such as fascicle lengths, age, sex, and previous injury history [24,31].

### 4.3. Limitations 

Although the NHE shows significant promise, the known studies still face some limitations that can impact the interpretation of the results and the generalizability of the conclusions. One of the main limitations is related to the variability in the methodologies adopted in the studies, which can jeopardize the consistency of the results and the comparison between different investigations. In addition, many studies do not take into account important external variables, such as the participants’ previous physical condition or consistency in the execution of the exercises. These factors can significantly influence the results and effectiveness of NHE in preventing injuries and improving athletic performance. It is therefore recommended that future research adopt a more standardized approach, both in the definition of training protocols and in the selection and assessment of participants. This also includes standardizing assessment methods in order to ensure consistency and comparability of results between studies. However, it is essential that future studies carefully consider external variables, such as a history of previous injuries, fitness level, and consistency in the execution of exercises, to identify possible confounding factors and improve the accuracy of the conclusions. In addition, the inclusion of larger and more heterogeneous participant samples will be important. This will allow for a more comprehensive and representative analysis of the effects of NHE in different populations and sporting contexts.

In short, to advance understanding of the role of NHE in injury prevention and athletic performance enhancement, it is crucial that future research addresses and mitigates the limitations identified by aiming for a more solid and conclusive investigation of the topic.

## 5. Conclusions

The integration of Nordic exercise into training programs is highly recommended, based on substantial evidence of its benefits for performance and injury prevention. However, in order to maximize its benefits, it is crucial that sports and clinical practitioners tailor training programs to meet the specific needs of the individuals. 

## Figures and Tables

**Figure 1 healthcare-12-01462-f001:**
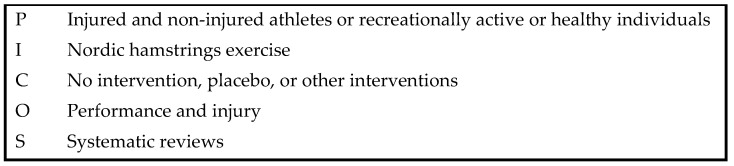
Description of the terms used to guide search strategy using the PICOS model: P—Patients; I—Intervention; C—Comparison; O—Outcomes; S—Studies.

**Figure 2 healthcare-12-01462-f002:**
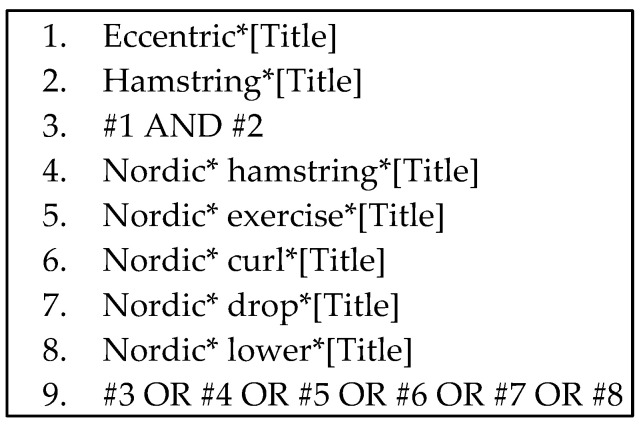
Description of the online search strategy.

**Figure 3 healthcare-12-01462-f003:**
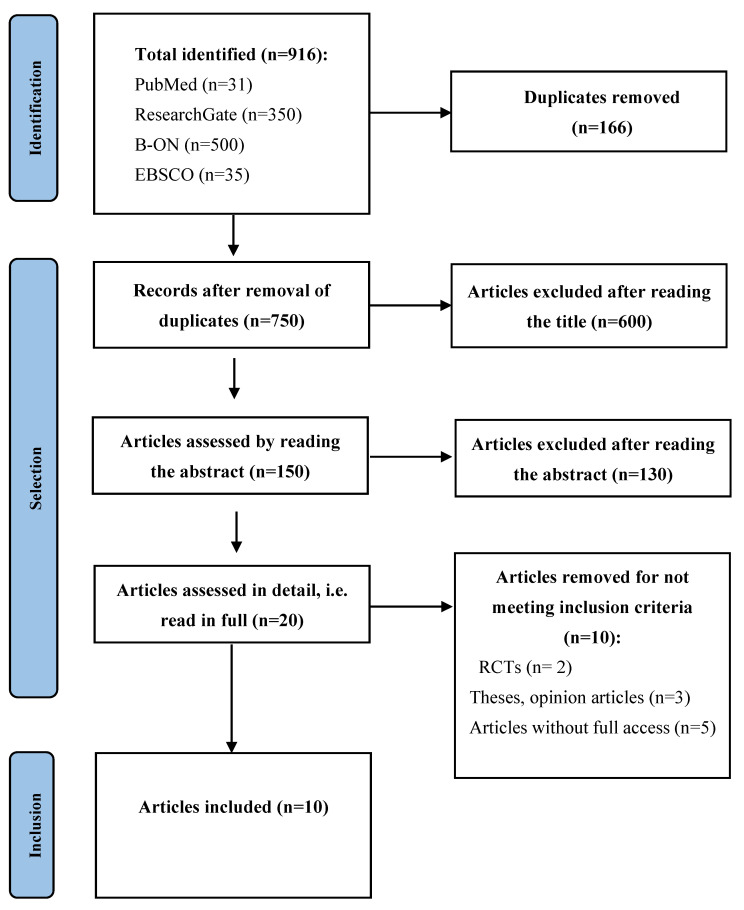
Study’s flow diagram. **Abbreviations:** RCTs, Randomized Controlled Trials.

**Table 1 healthcare-12-01462-t001:** Inclusion and exclusion criteria.

Inclusion The Studies Must	Exclusion The Studies Cannot
have at least one the keywords;	
be systematic reviews (with or without meta-analysis), prior to January 2024;	be books, case reports, expert opinions, conference papers, academic thesis, literature reviews or narrative reviews;
be published in peer-review journals;	
include injured and non-injured athletes or recreationally active or healthy individuals;	include experimental or control groups composed of any kind of animal;

**Table 2 healthcare-12-01462-t002:** Methodological quality of eligible studies.

Studies (A to Z)	AMSTAR 2-Items																AMSTAR-2 Score
	1	2	3	4	5	6	7	8	9	10	11	12	13	14	15	16	
Al Attar et. al., 2017 [24]	Yes	Yes	No	Partially yes	Yes	Yes	No	Yes	Yes	No	Yes	No	No	No	No	Yes	Critically Low
Bautista et al., 2021 [25]	Yes	No	Yes	Partially yes	No	No	No	Yes	Yes	No	Yes	No	Yes	No	Yes	Yes	Low
Cuthbert et al., 2020 [26]	No	No	No	Partially yes	No	No	No	Yes	Yes	No	Yes	No	No	No	No	Yes	Critically Low
Gérard et al., 2020 [27]	Yes	Yes	No	Partially yes	Yes	Yes	No	Yes	Yes	No	Yes	No	No	No	No	No	Critically Low
Hibbert et al., 2008 [28]	Yes	Yes	No	Partially yes	Yes	Yes	No	Yes	Partially yes	No	No meta-analysis was conducted	No meta-analysis was conducted	No	No	No meta-analysis was conducted	No	Critically Low
Llurda-Almuzara et al., 2021 [29]	Yes	Yes	No	Partially yes	Yes	No	No	Partially yes	Yes	No	No meta-analysis was conducted	No meta-analysis was conducted	No	No	No meta-analysis was conducted	Yes	Low
Medeiros et al., 2021 [30]	No	Yes	No	Partially yes	Yes	Yes	No	Partially yes	Yes	No	Yes	No	No	No	No	No	Critically Low
Opar et al., 2021 [31]	Yes	Yes	Yes	Partially yes	Yes	Yes	No	Yes	Yes	No	Yes	No	Yes	Yes	Yes	Yes	Moderate
Van Dyk et al., 2019 [32]	Yes	Yes	No	Partially yes	Yes	Yes	No	Partially yes	Yes	No	Yes	No	No	No	Yes	Yes	Low
Yagiz et al., 2021 [33]	No	No	No	Partially yes	Yes	Yes	No	Partially yes	Yes	No	Yes	No	No	No	Yes	Yes	Critically Low

**AMSTAR-2 items**: 1—Did the research questions and inclusion criteria for the review include the components of PICOS?; 2—Did the report of the review contain an explicit statement that the review methods were established prior to the conduct of the review and did the report justify any significant deviations from the protocol?; 3—Did the review authors explain their selection of the study designs for inclusion in the review?; 4—Did the review authors use a comprehensive literature search strategy?; 5—Did the review authors perform study selection in duplicate?; 6—Did the review authors perform data extraction in duplicate?; 7—Did the review authors provide a list of excluded studies and justify the exclusions?; 8—Did the review authors describe the included studies in adequate detail?; 9—Did the review authors use a satisfactory technique for assessing the risk of bias (RoB) in individual studies that were included in the review?; 10—Did the review authors report on the sources of funding for the studies included in the review?; 11—If meta-analysis was performed, did the review authors use appropriate methods for statistical combination of results?;12—If meta-analysis was performed, did the review authors assess the potential impact of the RoB in individual studies on the results of the meta-analysis or other evidence synthesis? 13—Did the review authors account for the RoB in individual studies when interpreting/discussing the results of the review?; 14—Did the review authors provide a satisfactory explanation for, and discussion of, any heterogeneity observed in the results of the review?; 15—If they performed quantitative synthesis, did the review authors carry out an adequate investigation of publication bias (small study bias) and discuss its likely impact on the results of the review?; 16—Did the review authors report any potential sources of conflict of interest, including any funding they received for conducting the review?

**Table 3 healthcare-12-01462-t003:** Systematic reviews summaries.

Study	Objective	Nº of Studies (Subjects and Characteristics)	Results and Conclusions
** *Injuries* **			
Al Attar et al., 2017 [24]	Investigate the effectiveness of the injury prevention programs that included the NHE on reducing hamstring injury rates while factoring in athlete workload.	5 *n* = 4455 Population: Athletes (soccer) Age: 13–40 years Sex: 80% males	The pooled results showed 51% overall injury reduction per 1000 h of exposure in the NHE injury prevention program group compared to the control group (IRR 0.490; 95% CI 0.291 to 0.827; *p* = 0.008).
Hibbert et al., 2008 [28]	Determine the effectiveness of eccentric exercise in preventing hamstring strains.	7 *n* = 818 Population: Athletes (soccer, rugby, Australian football) Age: 18–36 years Sex: 100% males	The cohort studies showed a lower incidence of hamstring strains with NHE, but no significant difference in severity of injury. The RCT found that the NHE did not decrease the risk of hamstring strains; however, participants who completed at least two training sessions sustained fewer hamstring injuries.
Opar et al., 2021 [31]	Identify the association between pre-season eccentric knee flexor strength quantified during performance of the NHE and the occurrence of future hamstring strain injury.	6 *n* = 1100 Population: Athletes (soccer, Australian football, Gaelic Football) Age: 25 ± 4 years Sex: 100% males	No significant differences in absolute knee flexor strength were observed between the prospectively injured limbs and the uninjured control group (SMD −0.22; 95% CI −0.50 to 0.05) or the recurrent injured limbs compared to the uninjured control group (SMD −0.32; 95% CI −0.77 to 0.13). Additionally, no significant differences in between-limb knee flexor strength asymmetry were found between all injured participants (SMD 0.01; 95% CI −0.24 to 0.25) or recurrently injured participants (SMD 0.28; 95% CI −0.14 to 0.70) compared to the uninjured group. Normalizing knee flexor strength to body mass had no effect on any outcome, and the pooled effect sizes were almost identical to the absolute knee flexor strength. Specifically, the effect size remained small for all injured limbs (SMD −0.23; 95% CI −0.55 to 0.10) or recurrently injured limbs (SMD −0.32; 95% CI −0.90 to 0.26) when compared to the uninjured group. The meta-regression did not show significant relationships between absolute knee flexor strength and any covariate investigated (sport played; athlete age, height, and mass; or average absolute NHE strength of cohort) were found (*p* ≥ 0.26). For between-limb asymmetry, a significant effect was found for average age (*p* = 0.007), but not any other variable (*p* ≥ 0.24).
van Dyk et al., 2019 [32]	Understand if the NHE prevents hamstring injuries when included as part of an injury prevention intervention.	15 *n* = 8459 Population: Athletes (soccer, rugby, baseball, and Australian football) Age: 13–40 years Sex: 87% males	There is a reduction in the overall injury risk ratio of 0.49 (95% CI 0.32 to 0.74; *p* = 0.0008) in favor of programs including the NHE. When pooling the RCTs, a small increase in the overall injury risk ratio of 0.52 (95% CI 0.32 to 0.85; *p* = 0.0008), still favors the NHE. Additionally, when studies with a high risk of bias were removed, there is an increase of 0.06 in the risk ratio to 0.55 (95% CI 0.34 to 0.89; *p* = 0.006).
** *Performance* **			
Bautista et al., 2021 [25]	To investigate the effects of the NHE on sprint performance (i.e., 5, 10, and 20 m) and explore associations between study characteristics and sprint outcomes in team sport players.	20 *n* = 506 Population: Recreationally active individuals or Athletes (soccer, rugby, sprinters, and Australian football) Age: 13–40 years Sex: 86% males	NHE interventions showed a positive effect on sprint performance (MD −0.04 s; 95% CI −0.08 to −0.01). Sub-group meta-analyses indicated no significant differences in 5 (MD −0.02 s; 95% CI −0.10 to 0.06) and 20 m (MD −0.05 s; 95% CI −0.30 to 0.19) sprint performance. A significant difference was however found for 10 m sprint performance (MD −0.06 s; 95% CI −0.10 to −0.01). Meta-analysis on the effects of the NHE on eccentric strength of the knee flexors showed a significant benefit in favor of the intervention group (SCMD 0.83; 95% CI 0.55 to 1.12).
Cuthbert et al., 2020 [26]	To investigate the effect of NHE-training volume on eccentric hamstring strength and biceps femoris fascicle length adaptations.	13 *n* = 589 Population: Recreationally active individuals, Physically active individuals, or Athletes (soccer, handball, basketball) Age: 10–30 years Sex: 82% males	Within-study differences showed that following interventions of ≥ 6 weeks, very large positive effect sizes were seen in eccentric strength following both high-volume (g = 2.12) and low-volume (g = 2.28) NHE interventions. Similar results were reported for changes in fascicle length (g ≥ 2.58) and a large-to-very large positive reduction in pennation angle (g ≥ 1.31). Between-study differences were estimated to be at a magnitude of 0.374 (*p* = 0.009) for strength and 0.793 (*p* < 0.001) for architecture.
Gérard et al., 2020 [27]	To determine the effects of an eccentric hamstring strength-training program, performed for at least 4 weeks by healthy adults, on muscle architecture and eccentric strength	10 *n* = 346 Population: Recreationally active individuals or Athletes (soccer) Age: 18–30 years Sex: 100% males	Eccentric strength training was associated with an increase in fascicle length (MD 1.97; 95% CI 1.48 to 2.46) and muscle thickness (MD 0.10; 95% CI 0.06 to 0.13), and a decrease in pennation angle (MD 2.36; 95% CI 3.11 to 1.61). An increase was also found after eccentric strength training compared with concentric strength training (SMD 1.06; 95% CI 0.26 to 1.86), usual level of activity (SMD 2.72; 95% CI 1.68 to 3.77), and static stretching (SMD 0.39; 95% CI −0.97 to 1.75).
Llurda-Almuzara et al., 2021 [29]	To evaluate the biceps femoris long head activation across cross-sectional hamstring strength exercise studies.	29 *n* = 507 Population: Healthy adults without lower extremity injuries Age: 19–30 years	Isokinetic and NHE as the categories with highest biceps femoris activation (>60% of MVIC). NHE ankle dorsiflexion was the exercise that achieved the highest biceps femoris long head activation (128.1% of its MVIC).
Medeiros et al., 2021 [30]	To investigate the effects of NHE on knee flexor eccentric strength and fascicle length.	12 *n* = 299 Population: Recreationally active individuals, Physically active individuals, or Athletes (soccer, hockey) Age: 18–30 years Sex: 74% males	The studies demonstrated strength increments in response to NHE (10–15% and 16–26% in tests performed on the isokinetic dynamometer and on the NHE device, respectively), as well as significant enhancement of biceps femoris long head fascicle length (12–22%). Meta-analysis showed NHE training was effective in increasing knee flexor eccentric strength, assessed with both isokinetic tests (SMD 0.68; 95% CI 0.29 to 1.06) and NHE tests (SMD 1.11; 95% CI 0.62 to 1.61). NHE was also effective in increasing fascicle length (SMD 0.97; 95% CI 0.46 to 1.48).
Yagiz et al., 2021 [33]	To explore the effects of eccentric training based on biceps femoris fascicle length using ultrasound assessment and extrapolation methods.	8 *n* = 181 Population: Recreationally active individuals, Physically active individuals, or Athletes (soccer) Age: 18–30 years	NHE showed small (g = 0.23; 95% CI −1.02 to 1.47), medium (g = 0.38; 95% CI −0.50 to 1.27) and large (g = 1.98; 95% CI 0.52 to 3.44) ES based on manual linear extrapolation, panoramic ultrasound scanning, and trigonometric equation methods, respectively.

**Abbreviations**: CI, Confidence Intervals; ES, Effect Size; g, Hedges’ g; IRR, Incidence Rate Ratio; MD, Mean Differences; MVIC, Maximal Voluntary Isometric Contraction; NHE, Nordic Hamstring Exercise; p, probability value; SCMD: Standardized Change of Mean Difference; SMD; Standardized Mean Differences.

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
