# Peer review of "The Effects of Nordic Hamstring Exercise on Performance and Injury in the Lower Extremities: An Umbrella Review"

_healthcare, 2024, doi:10.3390/healthcare12151462_

Round 1

Reviewer 1 Report

Comments and Suggestions for Authors

The aim of this umbrella review was to summarize the effects of Nordic Hamstring Exercise (NHE) on performance and injury prevention. This study adhered to the PRISMA guidelines, conducting a comprehensive literature search across multiple e-databases, following the P (injured and non-injured athletes or recreationally active or healthy individuals) I (NHE) C (no intervention, placebo, or other interventions) O (performance or injury) S (systematic reviews) model. This article is well-written, especially in the discussion section. However, with some revisions in the introduction, it could be suitable for publication in the journal.

L1-3: It is suggested that the title be revised with one of the following recommendations:

  1. Include general characteristics of the subjects.
  2. Add "in the lower extremity" to the injury description.
  3. Specify which aspects of performance were primarily considered by the researchers.

L24-25: The conclusion in the abstract resembles the results section. It is suggested that the authors include a practical takeaway statement in this part.

L39: What is meant by "In the literature, there are several interventions that are intended to strengthen and reduce the risk of injury (such as strain), through exercises that are able to enhance the hamstring muscle, such as Nordic hamstring exercise (NHE) [3-6]."? What aspect of the muscle is being enhanced? The sentence needs to be revised.

L46: You have stated "A greater superiority of eccentric training compared to concentric training is due...". How does the type of contraction relate to the objectives of this study?

L52: Was there any inconsistency in the findings of previous studies? In the introduction, the problem statement and the necessity of the research were not properly articulated.

Author Response

Thank you for your care in reviewing our study. As recommended, we will respond to the comments point-by-point.

Comment 1: L1-3: It is suggested that the title be revised with one of the following recommendations:

  1. Include general characteristics of the subjects.
  2. Add "in the lower extremity" to the injury description.
  3. Specify which aspects of performance were primarily considered by the researchers.

Response 1: Thank you for your recommendation. We think option 2 is the most suitable for the proposal. Therefore, we added the text as requested.

Comment 2: L24-25: The conclusion in the abstract resembles the results section. It is suggested that the authors include a practical takeaway statement in this part.

Response 2: Thank you for your suggestion. We changed the abstract conclusion.

Comment 3: L39: What is meant by "In the literature, there are several interventions that are intended to strengthen and reduce the risk of injury (such as strain), through exercises that are able to enhance the hamstring muscle, such as Nordic hamstring exercise (NHE) [3-6]."? What aspect of the muscle is being enhanced? The sentence needs to be revised.

Response 3: You are right, the sentence was confusing. We changed accordingly.

Comment 4: L46: You have stated "A greater superiority of eccentric training compared to concentric training is due...". How does the type of contraction relate to the objectives of this study?

Response 4: This sentence was for making a statement about the advantages of doing eccentric training over, for example, concentric training. As the NHE is an eccentric exercise, we think that the introduction has to justify the importance of the contraction type. Therefore, we added that paragraph. Nevertheless, we changed the paragraph for a better understanding.

Comment 5: L52: Was there any inconsistency in the findings of previous studies? In the introduction, the problem statement and the necessity of the research were not properly articulated.

Response 5: Thank you for your comment. Indeed, while a definitive consensus could not be established for most findings, the results from the systematic review generally align with and complement each other. Consequently, our primary motivation for conducting this study was to be the first to perform an umbrella review addressing the main outcomes. To fulfill the request, a clarifying statement has been added to address this issue.

Reviewer 2 Report

Comments and Suggestions for Authors

To the authors: Thanks you for preparing this review.  Your selection of appropriate manuscripts to include allowed what I would call a thorough examination of Nordic Hamstring Exercise even if was not called NHE. The umbrella approach benefits the reader for the organization of your manuscript can be searched for specific results. Headings for the Results and Discussion (1. Performance and 2. Injuries) are appreciated. That you itemized the highlighted for each study selected is a great manuscript structure.  Be sure to have a Title for Figure 3 and list any abbreviations.   

Author Response

Thank you for your care in reviewing our study.

Comment 1: Be sure to have a Title for Figure 3 and list any abbreviations.

Response 1: Thank you for your observation. As requested, we added the title and abbreviation for Figure 3

Reviewer 3 Report

Comments and Suggestions for Authors

First of all, I would like to congratulate the authors for their enormous effort. These data are of relevant importance for physical trainers and physiotherapists.

The work is well organized and well written. A few suggestions are made below to improve the work.

Issues

-The introduction provides a rationale for strengthening the hamstrings. However, in the introduction, there is no justification for carrying out this review. Please provide a justification for performing this review.

-Table 2 needs to be interpreted and discussed in theDiscussion section.For instance, in a topic calledStudy qualities.”

It is crucial to include a section titled 'Practical applications', where the authors can provide recommendations for interventions to prevent injury and enhance performance based on the studies presented in Table 3. I know that such recommendations were made along with the discussion, but creating such a topic will improve the paper reading.

-Reference 21 in the introduction did not support the following claims:NHE continues to cause some doubts within sport as to its applicability and reliability”.

-Table 3 needs to provide information about the population studied in each study.

-Line 61- insert ")"

-Line 72- "MeSH" instead mesh.

Author Response

Thank you for your care in reviewing our study. As recommended, we will respond to the comments point-by-point.

Comment 1: The introduction provides a rationale for strengthening the hamstrings. However, in the introduction, there is no justification for carrying out this review. Please provide a justification for performing this review.

Response 1: Thank you for your comment. As requested, text was added to clarify the issue.

Comment 2: -Table 2 needs to be interpreted and discussed in the “Discussion section.” For instance, in a topic called “Study qualities.”

Response 2: Thank you for your observation. While we understand your perspective on this issue, we kindly disagree with the necessity of adding a new sub-section titled "Study Qualities." In our study, the systematic reviews were evaluated using the AMSTAR-2 tool, with results presented in Table 2 and summarized in Section 3.2. This information helps assess the quality of the systematic reviews and, consequently, the weight of the recommendations and evidence. However, the importance of this may be lessened when systematic reviews are based on secondary studies (such as other systematic reviews), which are also influenced by their primary studies. Therefore, in our study (in the discussion section), we focused on summarizing the evidence using the best information weighting the evidence of the systematic reviews, while providing more specific orientations based on their primary studies (RCTs).

Comment 3: It is crucial to include a section titled 'Practical applications', where the authors can provide recommendations for interventions to prevent injury and enhance performance based on the studies presented in Table 3. I know that such recommendations were made along with the discussion, but creating such a topic will improve the paper reading.

Response 3: Thank you for the opportunity to clarify this point. As you noticed, we tried to include practical applications in the performance and injury sub-sections, wherever possible. Initially, we considered adding such sub-section (practical applications), after the injury sub-section, to provide guidance based on our findings. However, during the construction of our study, we realized that this would not only increase its length (which is already 16 pages) but might also result in some repetition of information already provided in the previous sections. Therefore, if you agree, we prefer to maintain the current structure of the study.

Comment 4: -Reference 21 in the introduction did not support the following claims: “NHE continues to cause some doubts within sport as to its applicability and reliability”.

Response 4: You are totally right. The reference numbered 21 was intended to be added in the first citation within the phrase. Therefore, we changed accordingly.

Comment 5: -Table 3 needs to provide information about the population studied in each study.

Response 5: Thank you for your observation. We constructed our Table 3, following the same structure of Table 2 from the Aromataris et al. 2015 study (for more information see: Aromataris, E., Fernandez, R., Godfrey, C. M., Holly, C., Khalil, H., & Tungpunkom, P. (2015). Summarizing systematic reviews: methodological development, conduct and reporting of an umbrella review approach. JBI Evidence Implementation13(3), 132-140.). We think that this information was most suitable in text format, as explored in the 3.3 sub-section. However, if you believe that additional information about the studied population should be provided, we kindly request that you specify the exact data that is needed.

Comment 6: -Line 61- insert ")"

Response 6: Thank you for noticing that detail. As recommended, the “)” has been added.

Comment 7: -Line 72- "MeSH" instead mesh.

Response 7: Thank you for noticing that detail. As recommended, the term has been changed.

Round 2

Reviewer 1 Report

Comments and Suggestions for Authors

Thank you very much for your great effort to make the paper improved.

I have no further comment on this paper.

Author Response

Thank you once again for your care in reviewing our study

Reviewer 2 Report

Comments and Suggestions for Authors

The edits have improved the abstract and introduction. 

Author Response

(The authors gave the same response as above.)

Reviewer 3 Report

Comments and Suggestions for Authors

The authors have satisfactorily answered all the questions. Below is a minor issue for improving the article: the addition of information on the study samples. 

Response 5: “However, if you believe that additional information about the studied population should be provided, we kindly request that you specify the exact data that is needed.”

I mean the characteristics of the sample, for example, athletes from which sport and which age group. This already provides important information for interpreting Table 3. 

Author Response

Thank you for your care and the reply.

As requested the population's main characteristics were included.